# Molecular Typing of *Neisseria gonorrhoeae* Clinical Isolates in Russia, 2018–2019: A Link Between *penA* Alleles and NG-MAST Types

**DOI:** 10.3390/pathogens9110941

**Published:** 2020-11-12

**Authors:** Ilya Kandinov, Ekaterina Dementieva, Dmitry Kravtsov, Alexander Chestkov, Alexey Kubanov, Victoria Solomka, Dmitry Deryabin, Dmitry Gryadunov, Boris Shaskolskiy

**Affiliations:** 1Center for Precision Genome Editing and Genetic Technologies for Biomedicine, Engelhardt Institute of Molecular Biology, Russian Academy of Sciences, 119991 Moscow, Russia; Ilya9622@gmail.com (I.K.); kdem@biochip.ru (E.D.); solo13.37@yandex.ru (D.K.); grad@biochip.ru (D.G.); 2State Research Center of Dermatovenerology and Cosmetology, Russian Ministry of Health, 107076 Moscow, Russia; chestkov@cnikvi.ru (A.C.); alex@cnikvi.ru (A.K.); solomka@cnikvi.ru (V.S.); dgderyabin@yandex.ru (D.D.)

**Keywords:** *Neisseria gonorrhoeae*, *penA* alleles, *Neisseria gonorrhoeae* multi-antigen sequence typing (NG-MAST), drug resistance, benzylpenicillin, ceftriaxone

## Abstract

This work aimed to study *penA* gene polymorphisms in clinical isolates of *Neisseria gonorrhoeae* collected in Russia in 2018–2019 and the contribution of the *penA* allele type to susceptibility to β-lactam antibiotics. A total of 182 isolates were analyzed. *penA* allele types were determined by sequencing, and the minimum inhibitory concentrations (MICs) of benzylpenicillin and ceftriaxone were measured. The influence of genetic factors on MICs was evaluated by regression analysis. All isolates were susceptible to ceftriaxone, and 40.1% of isolates were susceptible to penicillin. Eleven *penA* allele types were identified. The mosaic type XXXIV *penA* allele and the Gly120Lys substitution in PorB made the greatest contributions to increasing the ceftriaxone MIC; the presence of the *bla*_TEM_ plasmid, Gly120Asp, Ala121Gly/Asn substitutions in PorB, and the adenine deletion in the promoter region of the *mtrR* gene caused an increase in the penicillin MIC. Among 61 NG-MAST types identified, the most frequent were types 228, 807, 9486, 1993, and 6226. A link between *penA* alleles and *Neisseria gonorrhoeae* multi-antigen sequence typing (NG-MAST) types was established. Resistance to two groups of β-lactam antibiotics was associated with non-identical changes in *penA* alleles. To prevent the emergence of ceftriaxone resistance in Russia, NG-MAST genotyping must be supplemented with *penA* allele analysis.

## 1. Introduction

Sexually transmitted infections (STIs) are among the most dangerous and rapidly spreading infections. According to the WHO, more than 370 million STIs cases are recorded annually, of which 87 million are diseases caused by the pathogenic microorganism *Neisseria gonorrhoeae* [1]. The causative agent of gonococcal infection can activate various mechanisms of antibiotic resistance relatively quickly, leading to clinical ineffectiveness of the antimicrobial drugs used for treatment.

Benzylpenicillin was previously widely used to treat gonorrhea, but over the course of several decades, *N. gonorrhoeae* has acquired a powerful defense against this drug. Thus, benzylpenicillin was excluded from most treatment regimens in the 1960s and 1970s. However, the analysis of susceptibility to penicillins remains of scientific and practical interest. Another β-lactam antibiotic, the third-generation cephalosporin ceftriaxone, is currently the drug of choice for the treatment of gonococcal infection in the Russian Federation and most countries worldwide; thus, detecting ceftriaxone-resistant isolates and analyzing the resistance mechanisms are highly relevant.

*N. gonorrhoeae* isolates showing resistance to cephalosporins have been found in many countries worldwide [2,3,4,5,6,7,8,9,10]. NG-MAST type 1407, the most frequently identified sequence type (ST) in European countries, can develop resistance to third-generation cephalosporins via multiple mechanisms, including, in most cases, the presence of a mosaic allele of the *penA* gene encoding penicillin-binding protein 2 (PBP2), nucleotide substitutions in the *porB* gene encoding the porin protein PorB1b, impeding the cellular entry of the antibiotic, and a deletion in the promoter of the pump regulator *mtrR* gene, leading to increased antibiotic efflux [2,4,5,11,12].

Namely, the mosaic structure of the *penA* gene, which resulted from homologous recombination between commensal *Neisseria* species, such as *Neisseria perflava*, *Neisseria sicca*, *Neisseria polysaccharea*, *Neisseria cinerea*, and *Neisseria flavescens*, is found in numerous cephalosporin-resistant isolates [13,14,15]. A mosaic gene can contain multiple nucleotide substitutions that can lead to changes in the amino acid sequence of the protein. For example, mosaic allele 37.001 contains 230 nucleotide substitutions as compared with wild-type *penA* allele 100.001, which results in the change of 121 amino acids (https://ngstar.canada.ca/alleles/penA).

A semi-mosaic structure has also been distinguished, which, in most cases, has fewer amino acid substitutions at the *N*-terminal end of the protein sequence (residues 1–214) than does the mosaic allele [16]. Mutation in mosaic genes affecting cephalosporin resistance involves Ala311Val, Ile312Met, Val316Thr/Pro/Ser, Ala501Pro/Val, Asn512Tyr, and Gly545Ser. In addition, mutations have been identified in non-mosaic *penA* alleles that increase the minimum inhibitory concentration (MIC) of third-generation cephalosporins: Ala501Val/Thr substitutions and Gly542Ser and Pro551Ser/Leu amino acid alterations [12,17,18]. Currently, *penA* alleles are usually numbered with Roman numerals from I to XXXVIII and further in Arabic numerals [16,19] (https://ngstar.canada.ca).

The Russian population of the causative agent of gonorrhea continues to demonstrate high resistance to penicillin, but no isolates resistant to ceftriaxone have been found to date [20,21]. This can be explained by the relative isolation of the Russian population of gonococci, as confirmed by the presence of sequence types (STs) and genogroups unique to the Russian Federation [22]. However, the evolution of drug resistance mechanisms and the transboundary transfer of globally widespread *N. gonorrhoeae* clones with established resistance mechanisms create a permanent risk of the emergence and spread of *N. gonorrhoeae* strains resistant to third-generation cephalosporins.

The goals of our work were to study *penA* gene polymorphisms in *N. gonorrhoeae* clinical isolates obtained in the Russian Federation in 2018–2019, to determine the contribution of the *penA* allele type to phenotypic resistance to β-lactam antibiotics (benzylpenicillin and ceftriaxone), to perform molecular typing of clinical isolates and to analyze the association between the *penA* allele type and gonococcal phylogeny as assessed by NG-MAST.

## 2. Results

### 2.1. Susceptibility of Isolates Collected in Russia in 2018–2019 to β-Lactam Antibiotics

A total of 182 *N. gonorrhoeae* clinical isolates collected in various regions of the Russian Federation in 2018–2019 were analyzed. All samples of the analyzed set were susceptible to ceftriaxone. However, two isolates with decreased sensitivity to ceftriaxone (MIC_cro_ = 0.12 mg/L) were found—one from Arkhangelsk in 2018 and the second from Astrakhan in 2019. Seventy-three isolates (40.1%) were sensitive to penicillin (MIC_pen_ ≤ 0.06 mg/L), 93 (51.1%) were moderately resistant (MIC_pen_ = 0.12–1 mg/L), and 16 (8.8%) were penicillin-resistant with MIC_pen_ > 1 mg/L, including two isolates with an extremely high MIC (MIC_pen_ = 32 mg/L).

For all isolates, genetic determinants associated with resistance to ceftriaxone and penicillin were identified. The characteristics of the studied isolates, i.e., *penA* allele type; *porB,* and *tbpB* types; NG-MAST type; MIC_pen_; MIC_cro_; mutations in the *ponA*, *penA*, *porB*, and *mtrR* genes; and the presence of the *bla*_TEM_ plasmid, are given in Appendix A.

### 2.2. Diversity of penA Gene Alleles in the Modern Russian Population of N. gonorrhoeae

The types of *penA* alleles in the Russian population of *N. gonorrhoeae* were determined for the first time. Among the 182 clinical isolates, 11 different *penA* allele types were found: I, II, V, IX, XIII, XIV, XV, XVIII, XXII, XXXIV, and 44. Allele XXXIV was of the mosaic type. The amino acid sequences of the identified alleles are shown in Figure 1. Wild-type alleles, i.e., sequences of the gene encoding the PBP2 protein with GenBank accession number M3209, were not found. The most common *penA* allele types were I and XV, which were represented by 60 and 55 isolates, respectively. These types differ from the wild-type allele by only one amino acid. It is necessary to note the widespread occurrence of the Asp345a insertion, which reduces susceptibility to penicillins but does not affect resistance to cephalosporins [12,23,24,25]. This mutation was present in 121 of the 182 isolates (66.5%), i.e., in all isolates except for those carrying *penA* alleles of types XV and XXXIV (mosaic) (Figure 1).

In addition, a number of alleles were characterized by mutations located at the C-terminal end of the transpeptidase, such as Phe504Leu, Ala510Val, Ala516Gly, and Pro551Ser(Leu) (Figure 1), which can reduce the rate of antibiotic binding and prevent subsequent conformational changes in the PBP2 protein [17,26].

### 2.3. NG-MAST and Phylogenetic Analysis

Among the 182 isolates studied, 61 NG-MAST types were identified, indicating the high genetic diversity of pathogenic strains circulating in the Russian Federation. The most common molecular types were STs 228 (20 isolates, 11.0%), 807 (16 isolates, 8.8%), 9486 (13 isolates, 7.1%), 1993 (11 isolates, 6.0%), and 6226 (8 isolates, 4.4%). A large number (27) of single ST isolates (i.e., STs represented by only one isolate) was found, a typical finding for the Russian Federation. The obtained data were similar to those for the NG-MAST types of isolates collected in the Russian Federation in 2013–2018, when STs 807, 228, and 1993 were also the most common STs [22,27].

According to the results of phylogenetic analysis, the analyzed *N. gonorrhoeae* isolates were united into genogroups that included the closest NG-MAST types. Table 1 represents genogroups containing four or more isolates; thus, 146 out of 182 isolates belonged to the listed genogroups (the rest isolates were not attributed to any genogroup). Genogroup G807, the most widespread genogroup in the Russian Federation, largely corresponds to the European genogroup G51 containing isolates obtained in the EU countries in 2013 [28], and the Russian genogroup G1152 corresponds to the European genogroup G387. In 2013–2018, the Russian genogroup G1152 was one of the most frequently isolated in Russia [22]. In the present work, isolates with STs 5734, 17,017, and 17,532 belonging to this genogroup were found; however, isolates of NG-MAST type 1152 were not found. Five additional genogroups (G6226, G9486, G5042, G18948, and G14942) contained STs exclusively characteristic of the Russian population of *N. gonorrhoeae* and, therefore, could not be correlated with European or globally widespread isolates.

Among the samples collected in 2018–2019, there were six isolates of STs 3149, 5622, and 10,025, which belong to the pandemically significant European genogroup G1407. NG-MAST 3149 and 5622 differ from NG-MAST 1407 by one nucleotide and from NG-MAST 10,025 by two nucleotides (substitutions in *porB*). The isolates from Russia demonstrating decreased sensitivity to ceftriaxone MIC_cro_ = 0.12 mg/L (two isolates) were of ST 5622 and ST 10,025 and thus belonged to genogroup G1407.

### 2.4. Relationship between the penA Allele Type and NG-MAST Type

A maximum-likelihood phylogenetic tree (phylogram) was constructed for the *N. gonorrhoeae* isolates obtained in the Russian Federation in 2018–2019 (Figure 2). The main genogroups and *penA* allele types are shown on the phylogram. Analysis of the data revealed a link between the NG-MAST type and the *penA* allele type and, accordingly, between the genogroups, as determined by the NG-MAST type and *penA* allele type.

The presence of only one *penA* allele type within a genogroup was a typical feature. All isolates of NG-MAST 228 (20 isolates), NG-MAST 807 (16 isolates), and NG-MAST 9570 (5 isolates), and other isolates belonging to genogroup G807 had the type I *penA* allele, i.e., the entire G807 genogroup carried the type I *penA* allele. All isolates of genogroup G6226 possessed the type V *penA* allele, and all isolates belonging to genogroup G1993 had the *penA* allele. Isolates of three NG-MAST types (ST 3149, 5622, and 10,025) belonging to genogroup G1407 carried the mosaic type XXXIV allele. Thus, a functional dependence was observed when the list of STs (the list of genogroups according to NG-MAST type) was mapped onto the list of *penA* alleles. Inverse dependence was not observed. For example, *penA* allele type XV was found both in isolates of genogroup G1993 and in isolates of other genogroups (G9486 (13 isolates) and G5042 (7 isolates)), as well as in the isolates of NG-MAST 5042, 14940, 19648, and 19578 (Figure 2). As seen from the phylogram, isolates carrying *penA* alleles of the same type can belong to phylogenetically close STs and be located on tree branches that are far from each other (for example, type II and types V and XV).

### 2.5. Effect of the Genetic Profile on the MIC_cro_ and MIC_pen_

To systematize the data and study the effect of mutations on the MIC_cro_ and MIC_pen_ values, the isolates were divided into groups A-L according to their genetic profile considering the *penA* allele type and the presence of mutations in other genes associated with resistance to β-lactam antibiotics (Table 2). To check the correctness of grouping, a linear regression model was constructed for the MIC values using the groups established according to the genetic profiles of the 182 isolates as independent variables. Regression analysis showed that this grouping was statistically significant (Appendix A):

for MIC_cro_–adjusted R-squared 0.7254, *p* < 0.001;

for MIC_pen_–adjusted R-squared 0.7041, *p* < 0.001.

Group A contained all 55 isolates with *penA* allele XV, and group B included almost all isolates with *penA* allele I (59 of 60 isolates). Group C consisted of isolates with *penA* type XXII. Some isolates in group C had a mutation in the *ponA* gene leading to Leu421Pro substitution; thus, group C was divided into subgroups C1 and C2. Isolates with *penA* allele type V were found in three groups with different genetic profiles (groups D, H, and I). Group J included isolates carrying the mosaic type XXXIV *penA* allele, and two isolates with MIC_cro_ = 0.12 mg/L were classified into this group. Group L included two isolates carrying the *bla*_TEM-1_ plasmid determinant, the presence of which led to an increase in the MIC_pen_ to 32 mg/L. I median MIC values were calculated for each group (groups A–L). Ten of the 182 isolates had unique genetic profiles that did not allow their inclusion in the selected groups (denoted as “ungrouped” in Table 2).

Although all isolates studied were susceptible to ceftriaxone, the MIC_cro_ values differed among the groups. The median value of MIC_cro_ values for other groups was increased compared with that for group A as follows:
−Up to 0.08 mg/L for the following genetic profiles: *penA* allele XXII, Leu421Pro replacement in PBP1 (*ponA* gene) (group C2); *penA* allele V, Leu421Pro replacement (*ponA* gene) or delA in the *mtrR* gene (groups D and I);−Up to 0.012–0.015 mg/L for the following genetic profiles: *penA* allele XIII, Leu421Pro replacement (*ponA* gene) (group E); *penA* allele V, Leu421Pro replacement plus mutations in the *porB* gene (group H); *penA* allele 44, Leu421Pro replacement plus delA in the *mtrR* gene, mutations in the *porB* gene (group K);−Up to 0.03 mg/L for the following genetic profile: *penA* allele IX, Leu421Pro replacement (*ponA* gene) plus mutations in the *porB* gene (group G);−Maximum increase in MIC_cro_ to 0.045 mg/L in group J, which contained isolates carrying the *penA* mosaic allele XXXIV, Leu421Pro replacement (*ponA* gene) plus delA in the *mtrR* gene, and mutations in the *porB* gene.

The isolates of all groups except group A showed resistance to penicillin (MIC_pen_ > 0.06 mg/L). Group B (MIC_pen_ = 0.12 mg/L) included isolates carrying the type I *penA* allele, which contains the insAsp345a mutation. Moderately resistant isolates with MIC_pen_ = 0.25–0.5 mg/L were included in groups C, D, and F; isolates with MIC_pen_ = 1.0–1.5 mg/L were included in groups G, I, K, H, and J. Finally, penicillin-resistant isolates with MIC_pen_ = 2 mg/L were included in group E and had the following genetic profile: *penA* allele XIII and Leu421Pro replacement (*ponA* gene). The isolates with MIC_pen_ = 32 mg/L carried the *bla*_TEM-1_ plasmid.

Changes in the MIC_cro_ and MIC_pen_ values by groups with different genetic profiles are shown as box plots in Figure 3 and Figure 4, respectively; groups A–L (172 isolates) for ceftriaxone and groups A–K (170 isolates) for benzylpenicillin (group L with isolates containing *bla*_TEM_ plasmid was excluded from the penicillin box plot on Figure 4). The box plots clearly demonstrate the increase in the MIC by group depending on the genetic profiles of the *N. gonorrhoeae* isolates.

### 2.6. Contribution of Genetic Determinants to the Resistance of N. gonorrhoeae Isolates to β-Lactam Antibiotics: Regression Analysis

Regression analysis was used to assess the effect of the analyzed molecular determinants on the resistance of *N. gonorrhoeae* isolates to β-lactam antibiotics. Regression models predicting the base 2 logarithmic MIC_cro_ and MIC_pen_ values were constructed for all 182 isolates. The predictor variables were the *penA* allele type, the amino acid polymorphism in the protein encoded by *ponA* (Leu421Pro), mutations in codons 120 and 121 of *porB*, the delA mutation in the promoter region of the *mtrR* gene, and the presence of the *bla*_TEM_ plasmid. The results of regression analysis of the samples are presented in Appendix A.

According to the analysis results, the presence of the mosaic type XXXIV *penA* allele (*p* < 0.001) made the greatest contribution to the increase in MIC_cro_. The Gly120Lys substitution in the porin protein (*porB* gene) had a lesser effect. The presence of *penA* allele type XV consistently decreased the MIC_cro_ (*p* < 0.001, decrease in ceftriaxone resistance). The mutation of the promoter region of the *mtrR* gene did not significantly affect the MIC_cro_ for the studied isolates. Regarding penicillin resistance, the greatest contribution to the increase in the MIC_pen_ (*p* < 0.001) was made by the *bla*_TEM_ plasmid encoding β-lactamase, which cleaves penicillins but is inactive against cephalosporins. In addition, the Gly120Asp and Ala121Gly substitutions in the porin protein and mutations in the promoter region of the *mtrR* gene had large effects. The calculations indicated that the MIC_pen_ was also significantly increased in the presence of *penA* alleles of types I, II, V, IX, XIII, and XXII, an effect that can be explained by the presence of the Asp345a insertion in these alleles. The mosaic allele, type XXXIV, did not affect penicillin resistance in this set of isolates, as confirmed by statistical analysis.

Thus, analysis of groups of isolates differing in genetic profiles (Table 2) and analysis of regression models constructed with several molecular determinants of resistance as independent variables showed similar results for identifying the determinants with the most noticeable effects on the MIC_cro_ and MIC_pen_.

## 3. Discussion

In this work, molecular typing of clinical isolates of *N. gonorrhoeae* obtained in the Russian Federation in 2018–2019, evaluation of *penA* gene polymorphisms, and analysis of resistance to β-lactam antibiotics were carried out. According to the EUCAST criteria, all isolates were susceptible to ceftriaxone (MIC_cro_. ≤ 0.125 mg/L); however, two strains with an increase in the MIC of ceftriaxone to 0.12 mg/L were found. Only 40.1% of the Russian isolates were susceptible to penicillin (MIC_pen_ ≤ 0.06 mg/L). Thus, although these antibiotics are β-lactams with an inherent fundamental similarity of “targets” in the bacterial cell, the sensitivity of these isolates to these antibiotics was qualitatively different. Notably, this situation is typical not only in the Russian Federation but also in other countries.

To clarify the mechanism of resistance of *N. gonorrhoeae* to different groups of β-lactams, the *penA* allele type in isolates from the Russian population was determined for the first time, and this characterization was supplemented by analysis of the determinants in the *penA*, *ponA*, *mtrR*, and *porB* genes. Eleven different types of *penA* alleles were found, one of which was of mosaic type XXXIV. The type I and XV *penA* alleles were the most common in the Russian Federation. Although all studied isolates were susceptible to ceftriaxone, mutations in the *penA*, *ponA*, *mtrR*, and *porB* genes caused an increase in the MIC_cro_. A similar profile was described in [29] for the Canadian population of *N. gonorrhoeae*, which is sensitive to third-generation cephalosporins, i.e., the Canadian population is similar to the Russian population of *N. gonorrhoeae*.

Linear regression analysis was used to study the influence of *penA* allele type and other molecular determinants of resistance on the MIC_cro_ and MIC_pen_ values in detail. The application of this method to predict the MIC values of various antimicrobial drugs against *N. gonorrhoeae* has been described previously [30,31,32]. The main determinants affecting MIC_cro_ have been shown to be five amino acid substitutions in PBP2 Ala311Val, Ala501Pro/Thr/Val, Asn512Tyr, Ala516Gly, and Gly542Ser (numeration of amino acids Asn513Tyr, Ala517Gly, and Gly543Ser is given in the paper), mutation of the *mtrR* promoter, substitutions of the Gly120 residue in the PorB protein, and the Leu421Pro substitution in PBP1 [32]. Analysis of 182 Russian isolates showed that the presence of the mosaic type XXXIV *penA* allele and the Gly120Lys substitution in the PorB protein made the greatest contributions to the increase in the MIC_cro_. Regarding penicillin, the presence of the *bla*_TEM_ plasmid, the Gly120Asp and Ala121Gly/Asn substitutions in the porin protein, and the deletion of an adenine in the promoter region of the *mtrR* gene contributed to the increase in the MIC_pen_.

Thus, these results showed both partial similarity and differences in the molecular mechanisms leading to the increased resistance to different groups of β-lactams. The most significant differences were those in the *penA* alleles, a number of non-mosaic variants of which were found in isolates with a high MIC_pen_; in contrast, an increase in MIC_cro_ was possible only when the mosaic allele was acquired. On the other hand, isolates characterized by increased values of both MIC_cro_ and MIC_pen_ had mutations in components of antibiotic transport and efflux systems and the amino acid substitution in the target PBP1 protein, indicating that these determinants constitute a “platform” for the development of cross-resistance to several groups of β-lactam antibiotics.

The phylogeny of the Russian *N. gonorrhoeae* population was assessed on the basis of the NG-MAST results. A total of 61 NG-MAST types were identified among the isolates collected in 2018–2019. Analysis of the Russian *N. gonorrhoeae* population considering NG-MAST types and genogroups carried out in our previous studies led to the conclusion about the predominantly local character of its formation and evolution, and phylogenetic analysis showed the difference between the most common Russian STs and the European STs [20,21,27,33]. This tendency was maintained for the Russian *N. gonorrhoeae* population in 2018–2019; NG-MAST 228, 807, 9486, and 1993 remained the most widespread types and retained their dominant position for many years.

Phylogenetic analysis divided *N. gonorrhoeae* isolates into genogroups that united the closest NG-MAST types. We showed the functional dependence of the *penA* allele type on the NG-MAST type, i.e., isolates of the same ST carried the same *penA* allele type. However, the converse is not true since isolates with the same *penA* alleles were found to belong to different STs and genogroups. The constructed phylogenetic tree demonstrated that isolates carrying *penA* alleles of the same type could both belong to phylogenetically close STs and be located on tree branches that are far from each other.

Among the Russian isolates, no isolates of NG-MAST 1407, a pandemically significant and widespread type in Europe and worldwide, were found. However, six isolates belonging to the G1407 genogroup were found. In addition, phylogenetic analysis showed that all representatives of the G1407 genogroup possessed the mosaic type XXXIV *penA* allele, which confirmed the above-described functional dependence of the *penA* allele type on the NG-MAST type. In addition, these isolates carried the Leu421Pro mutation in the PBP (*ponA* gene); Gly120Lys and Ala121Asn alterations in the porin protein PorB; and the deletion mutation in the promoter region of the *mtrR* gene (encoding an efflux pump). These determinants additionally caused a gradual increase in the MIC_cro_. Notably, globally widespread strains of *N. gonorrhoeae* with multidrug resistance, including high resistance to ceftriaxone and cefixime, that were isolated in Europe (France and Spain) belonged to NG-MAST type 1407 and had both a mosaic *penA* allele of exactly type XXXIV and the Ala501Pro substitution in PBP2 [4,5]. Thus, the risks of the emergence and spread of cephalosporin resistance in the Russian population of *N. gonorrhoeae* are currently associated with the transboundary transfer of representatives of the G1407 genogroup.

## 4. Materials and Methods

### 4.1. Ethics Statement

According to the local Ethics Committees of the State Research Center of Dermatovenerology and Cosmetology of the Russian Ministry of Health, this research does not require formal ethical approval. All specimens used in this study were not connected to any personal information about the patients. Specifically, no ID by name or address was provided; i.e., all samples were anonymous samples.

### 4.2. Collection and Characterization of N. gonorrhoeae Isolates

Clinical isolates of *N. gonorrhoeae* (182 samples) were obtained by the State Research Center of Dermatovenerology and Cosmetology of the Russian Ministry of Health from 7 regions of the Russian Federation: Arkhangelsk, Astrakhan, Kaluga, Kazan, Omsk regions, Moscow, and the Chuvash Republic. Samples were collected by specialized medical organizations of dermatovenerological profiling, and each sample was collected from an individual patient. Sample collection, transportation, cultivation, phenotyping, and storage were performed according to a previously described protocol [20,21].

### 4.3. Testing of N. gonorrhoeae Susceptibility to Penicillin and Ceftriaxone

The minimum inhibitory concentrations (MICs) of benzylpenicillin and ceftriaxone were determined by the serial dilution method on chocolate agar. The breakpoints for susceptibility (S) and resistance (R) were established according to the European Committee on Antimicrobial Susceptibility Testing (EUCAST) guidelines (The European Committee on Antimicrobial Susceptibility Testing. Breakpoint tables for interpretation of MICs and zone diameters. Version 10.0, 2020, https://www.eucast.org/fileadmin/src/media/PDFs/EUCAST_files/Breakpoint_tables/v_10.0_Breakpoint_Tables.pdf): penicillin G (S ≤ 0.06 mg/L, R > 1.0 mg/L) and ceftriaxone (S ≤ 0.125 mg/L, R > 0.125 mg/L).

### 4.4. Identification of Genetic Determinants of Antimicrobial Resistance Using an Oligonucleotide Microarray

Genetic determinants of antimicrobial resistance were identified using an oligonucleotide hydrogel-based low-density microarray as described previously [34,35]. The analyzed determinants associated with resistance to penicillin G [12,23] were as follows: mutations in the *ponA* gene resulting in the amino acid substitution Leu421Pro in penicillin-binding protein 1 (PBP1), mutations in the *penA* gene resulting in the insertion of an aspartate in the 345 position (insAsp345a) of PBP2, and the presence of *bla*_TEM_ plasmids encoding β-lactamases capable of hydrolyzing amide bonds in penicillins. The determinants associated with resistance to both penicillin and ceftriaxone [12,23,36,37] were as follows: mutations in the *porB* gene leading to the amino acid alterations Gly120Lys/Asp/Asn/Thr and Ala121/Asp/Asn/Gly/Ser in the porin protein, which decreased the permeability of the cell membrane; and deletion of the adenine (delA) at nucleotide position −35 or insertion of one or two thymidines (insT or insTT) at nucleotide position −10 in the promoter region of the *mtrR* gene, which increased the expression of the MtrCDE efflux pump.

### 4.5. Evaluation of PenA Gene Polymorphisms

The *penA* gene sequences in *N. gonorrhoeae* samples were determined by Sanger sequencing. Before sequencing, the *penA* gene was amplified using the following primers: 5′-GCCAAAGGGCTTAACTTGCT-3′ (forward) and 5′-CTACGATCCCAACAGACCCG-3′ (reverse). The obtained nucleotide sequences were aligned and compared with the reference sequence of the wild-type *penA* gene (GenBank accession number M32091), and the *penA* allele type was determined according to the nomenclature described in the literature [16,19].

The evolutionary history of *penA* genes was inferred by using the maximum likelihood method based on the Tamura-Nei model [38]. Evolutionary analyses were conducted in MEGA X [39].

### 4.6. NG-MAST and Definition of Genogroups

Molecular typing of *N. gonorrhoeae* isolates was performed using the standard NG-MAST protocol [40]. The variable internal regions of the *porB* and *tbpB* genes were amplified via PCR, and the subsequent products were purified and sequenced in a 3730xl Genetic Analyzer (Applied Biosystems, USA). Both the leading and reverse strands were assessed, and the sequencing data were processed using 3730/3730xl Data Collection Software v. 3.0. Allele numbers for the *porB* and *tbpB* sequences and sequence types (STs) were assigned via the NG-MAST database (www.ng-mast.net). In this paper, NG-MAST types were also named as STs.

A phylogenetically related genogroup was defined, as described in [28], as the set of STs (two alleles) for which the concatenated sequence of both alleles (880 bp) displayed 99.4% (875 bp) similarity to the concatenated sequence of both alleles of the main ST in the genogroup. As a rule, genogroups were named after the most frequently identified ST in the group, i.e., G807 was named after NG-MAST 807.

### 4.7. Construction of the Phylogenetic Tree

For each NG-MAST type, the *porB* and *tbpB* sequences were concatenated and imported into the alignment tool. A maximum-likelihood phylogenetic tree was generated using RAxML software, version 7.4.8 [41] (http://galaxy-dev.cnsi.ucsb.edu/osiris) with 500 rapid bootstrap inferences.

### 4.8. Statistical Analysis

Linear regression models predicting the MIC_pen_ and MIC_cro_ values (base 2 logarithm of the MIC, log_2_(MIC)) for the data on *N. gonorrhoeae* isolates were constructed in the stats version 3.6.2 package for R software using the following parameters as predictor variables: types of *penA* alleles, mutations in proteins encoded by the *ponA* or *porB* genes, mutations in the *mtrR* gene (promoter region), and the presence of the *bla*_TEM_ plasmid. The final decision for the selection of variables to include in the model was made as a result of the successive deletion of variables using the Akaike information criterion (AIC). Regression analysis was also used to group the studied isolates according to their genetic profiles.

Box plots were constructed using the ggplot2 software package (https://www.rdocumentation.org/packages/ggplot2/versions/3.3.2). Each box indicates the median MIC value for a group of isolates with a specific genetic profile, the lower (25th percentile) and upper (75th percentile) quartiles, the interquartile range, and data outliers.

## 5. Conclusions

NG-MAST 228, 807, 9486, and 1993 remained the predominant STs of *N. gonorrhoeae* in the Russian Federation, and a large proportion of genogroups persisted from year to year, indicating a significant contribution of clonal reproduction to the population structure of the causative agent of gonococcal infection. For the first time, *penA* allele types were identified for the Russian population of *N. gonorrhoeae*. The link between the *penA* allele type and NG-MAST type and genogroup was shown, i.e., isolates of the same ST and genogroup carried the same *penA* alleles. Moreover, isolates with the same *penA* alleles could belong to different STs, including genetically distant STs, and genogroups.

The genogroups predominant in the Russian Federation did not show signs of the development of ceftriaxone resistance typical of representatives of the transboundary genogroup 1407 with a mosaic *penA* allele. Although no isolates of the pandemically significant NG-MAST 1407 were found in the Russian Federation in 2018–2019, six isolates belonging to genogroup G1407 were identified, two of which possessed the mosaic type XXXIV *penA* allele. Mutations in the *penA*, *mtrR*, and *porB* genes were shown to cause a gradual increase in the MIC_cro_. The mosaic type XXXIV *penA* allele and Gly120Lys substitution in the PorB protein made the greatest contributions to the increase in the MIC_cro_ for the Russian *N. gonorrhoeae* isolates. These data indicate that NG-MAST results must be supplemented by the determination of the *penA* allele type and other genetic determinants of resistance, which are important considerations for monitoring the emergence and spread of ceftriaxone-resistant strains.

## Figures and Tables

**Figure 1 pathogens-09-00941-f001:**
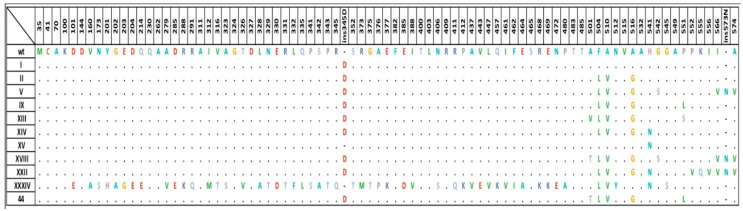
Amino acid sequences (according to the Ohnishi classification) of *penA* alleles identified in *Neisseria gonorrhoeae* isolates collected in the Russian Federation in 2018–2019 compared with the sequence of the wild-type allele (wt).

**Figure 2 pathogens-09-00941-f002:**
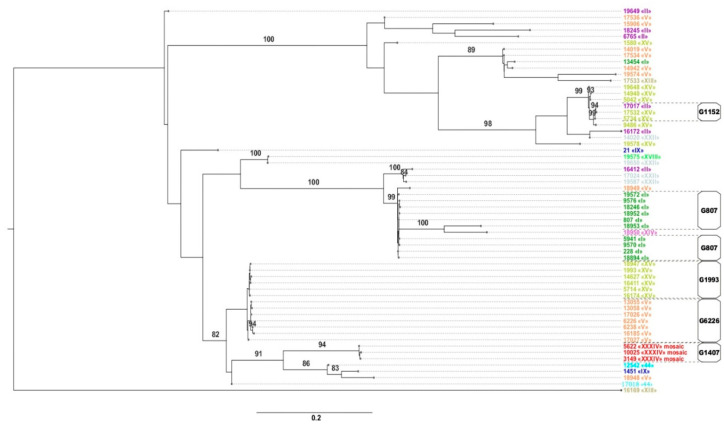
Phylogenetic tree constructed for the *N. gonorrhoeae* isolates obtained in the Russian Federation in 2018–2019. The branches indicate bootstrap values of ≥80. On the right, NG-MAST types and *penA* allele types are designated in different colors; in addition, isolates belonging to the main genogroups are indicated.

**Figure 3 pathogens-09-00941-f003:**
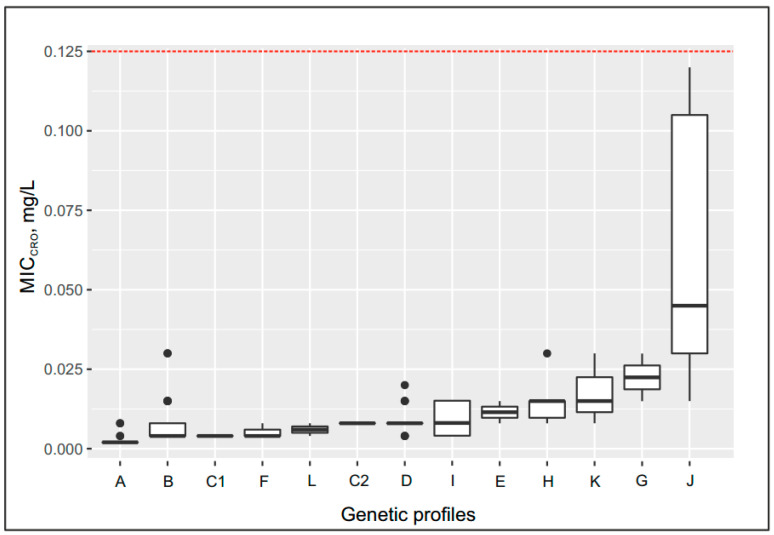
Box plot: Minimum inhibitory concentrations (MIC)_cro_–groups of *N. gonorrhoeae* isolates with different genetic profiles. A description of the genetic profiles by group is given in Table 2. The box indicates the first and third quartiles. Thick solid lines in each box indicate the median MIC_cro_ value for each group. The whiskers extend to the boundaries of the interquartile range. Data beyond the end of the whiskers are outliers and plotted as points. The Red line indicates the breakpoint for ceftriaxone resistance.

**Figure 4 pathogens-09-00941-f004:**
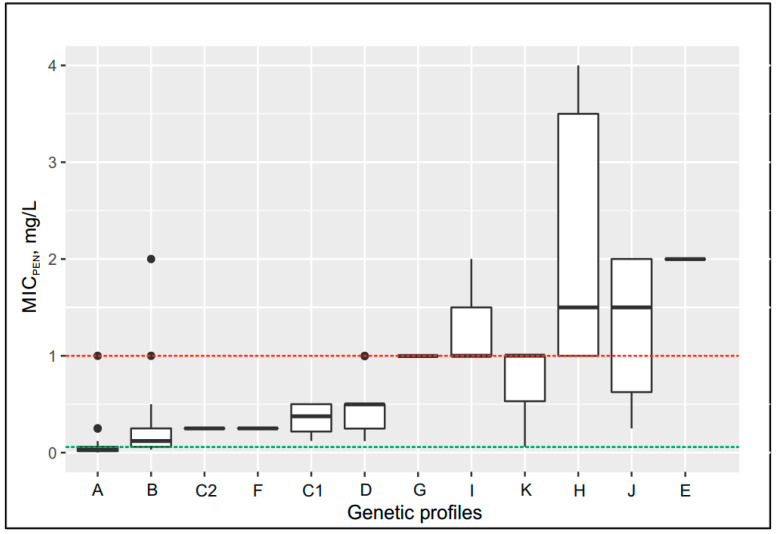
Box plot: MIC_pen_–groups of *N. gonorrhoeae* isolates with different genetic profiles. The designations on the boxes are the same as those in Figure 3. Thick solid lines in each box indicate the median MIC*_pen_* value for each group. Green and red lines indicate the breakpoints for penicillin resistance.

**Table 1 pathogens-09-00941-t001:** Main genogroups of the *N. gonorrhoeae* isolates collected in 2018–2019.

Genogroup	NG-MAST Types	Number of Isolates in the Genogroup	Notes
G807 *	228, 807, 5941, 9570, 9576, 18246, 18894, 18952, 19572	57	Corresponds to the European genogroup G51 [28]
G1993 *	1993, 5714, 14627, 16174, 16411, 17027, 18947	29	European genogroup G1993 [28]
G6226	6226, 6238, 13055, 13058, 16185, 17026, 17027	19	
G9486 *	9486	13	
G5042	5042,14940, 19648	7	
G1407 (European)	3149, 5622, 10025	6	
G18948	18948	6	
G14942 *	14942	5	
G1152 *	5734, 17017, 17532	4	Corresponds to the European genogroup G387 [28]

* Genogroups most frequently found in Russia in 2013–2018.

**Table 2 pathogens-09-00941-t002:** *penA* allele type and genetic profiles of *N. gonorrhoeae* isolates.

Group	Number of Isolates	NG-MAST	*penA*	MIC (mg/L), Group Median		Genetic Resistance Determinants				
				MIC_cro_	MIC_pen_	*ponA*	*mtrR*		*porB*	*bla* _TEM_
							−35	−10		
A	55	1580, 1993, 5042, 5714, 5734, 9486, 14627, 14940, 16174, 16411, 17532, 18947, 19578, 19648	XV	0.002	0.03	wt	wt	wt	wt	-
B	59	228, 807, 5941, 9570, 9576, 18246, 18894, 18952, 18953, 19572	I	0.004	0.12	wt	wt	wt	wt	-
C1	4	14020, 17024, 19587	XXII	0.004	0.375	wt	wt	wt	wt	-
C2	2	19650	XXII	0.008	0.25	Leu421Pro	wt	wt	wt	-
D	21	6226, 6238, 14019, 14942, 17026, 17027, 17536	V	0.008	0.5	Leu421Pro	wt	wt	wt	-
E	2	16169	XIII	0.012	2	Leu421Pro	wt	wt	wt	-
F	3	16172, 17017	II	0.004	0.25	Leu421Pro	wt	wt	wt	-
G	2	1451	IX	0.023	1	Leu421Pro	wt	wt	Gly120Asp	-
H	6	18948	V	0.015	1.5	Leu421Pro	wt	wt	Gly120Lys, Ala121Gly	-
I	7	13055, 13058, 16185, 17534, 18949	V	0.008	1	Leu421Pro	delA	wt	wt	-
J	6	3149, 5622, 10025	XXXIV (mosaic)	0.045	1.5	Leu421Pro	delA	wt	Gly120Lys/Asp, Ala121Asn/Gly	-
K	3	12542	44	0.015	1	Leu421Pro	delA	wt	Gly120Asp	-
L	2	18245, 18950	II, XIV	0.006	32	wt	wt	wt	wt	*bla*_TEM-1_, African type
Un- grouped	10	21, 6765, 13454, 15906, 16412, 17018, 17533, 19574, 19575, 19649	I, II, V, IX, XIII, XVIII, 44	0.004–0.03	0.12–4	wt or Leu421Pro	wt or delA	wt	wt or Gly120Lys/Asp, Ala121Asn/Asp	-

MIC-Minimum inhibitory concentration.

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
