# Peer review of "Molecular Typing of Neisseria gonorrhoeae Clinical Isolates in Russia, 2018–2019: A Link Between penA Alleles and NG-MAST Types"

_pathogens, 2020, doi:10.3390/pathogens9110941_

Round 1
Reviewer 1 Report
The manuscript "Molecular Typing of Neisseria gonorrhoeae Clinical Isolates in Russia, 2018-2019: a Link Between penA Alleles and NG-MAST Types" describes a molecular epidemiological study of Neisseria gonorrhoeae, isolated in Russia in 2018-2019. Authors present a well-designed study performed with standards enough for publishing in the journal. Since the manuscript was written by non-English-speaking authors, I would highly recommend English editing, I found some unusual chunks of language.
Although I did not found major problems with the manuscript, I present my comments and suggestions below, which help to improve the text.
Lines 7 and 9: Provide emails for all authors.
Line 27: Since the authors found more than one penA allele, I would recommend replacing them with alleles or allele types.
Line 45: Do not italicize a protein name.
Line 48: Repeat what was said above in the same sentence (line 45). Remove one of them.
Line 86: Could you add the region's information to Table S1? It would be interesting to know from which regions of Russia resistant strains are more often isolated.
Line 95: Space between "wide-spread" and "occurrence".
Line 100: I suggest making Figure 1 in color (amino acids).
Line 110: Is NG-MAST types and STs are synonyms? If so, choose one of them and use it so as not to confuse readers. Alternatively, justify in the Materials and Methods that they are the synonyms.
Line 114: Misprint: 1993, not 193.
Line 119: In reference [19] strains were isolated in 2013-2018, not 2018-2019 as in the text.
Line 125: What about other isolates? Are they have any genogroup? Describe in the text.
Line 125: "G1993* Corresponds to the European genogroup G1993 [25]". Is this the same genogroup, isn't it?
Line 125: "Two isolates from this genogroup demonstrated decreased sensitivity to ceftriaxone (MICcro = 0.12 mg/L)". The same repeated in the text of the manuscript (lines 130-132).
Line 125: "Corresponds to the European genogroup G387 [23]". Maybe reference 25?
Line 256: N.
Line 328: Fix the broken weblink.
Lines 360-364: Please, provide here the reference for the definition of genogroups.
Line 367: Change "RaxML" to "RAxML".
Line 491: Fix link to the working one: https://www.ecdc.europa.eu/sites/default/files/documents/Molecular-typing-N-gonorrhoeae-web.pdf
Author Response
Reviewer 1:
Since the manuscript was written by non-English-speaking authors, I would highly recommend English editing, I found some unusual chunks of language.
We agree that authors are non-native English writers and we use special proof-reading service. The manuscript was undergone the round of edition by American Journal Experts (www.aje.com).
Lines 7 and 9: Provide emails for all authors.
E-mails for all authors were added – lines 8-9 and 12-13.
Line 27: Since the authors found more than one penA allele, I would recommend replacing them with alleles or allele types.
“penA allele” was changed to “penA alleles” in the Keywords (line 31 (former line 27).
Line 45: Do not italicize a protein name;
Line 48: Repeat what was said above in the same sentence (line 45). Remove one of them.
The protein was named correctly and the repeat was removed. The text was changed to (Lines 46-52):
“N. gonorrhoeae isolates showing resistance to cephalosporins have been found in many countries worldwide [2–10]. NG-MAST type 1407, the most frequently identified sequence type (ST) in European countries, can develop resistance to third-generation cephalosporins via multiple mechanisms, including, in most cases, the presence of a mosaic allele of the penA gene encoding penicillin-binding protein 2 (PBP2), nucleotide substitutions in the porB gene encoding the porin protein PorB1b, impeding cellular entry of the antibiotic, and a deletion mutation in the promoter of the pump regulator mtrR gene, leading to increased antibiotic efflux [2,4,5,11,12].”
Line 86: Could you add the region's information to Table S1? It would be interesting to know from which regions of Russia resistant strains are more often isolated.
The column “Region of isolation” was added to Table S1 (Supplementary material).
Two isolates with decreased sensitivity to ceftriaxone (MICcro = 0.12 mg/L) were isolated, one in Arkhangelsk from 2018 and the second from Astrakhan in 2019 (this is mentioned in the text, lines 85-86).
Line 95: Space between "wide-spread" and "occurrence".
The space was added (Line 102 (former line 95).
Line 100: I suggest making Figure 1 in color (amino acids).
Figure 1 in black and white was changed for colored Figure 1.
Line 110: Is NG-MAST types and STs are synonyms? If so, choose one of them and use it so as not to confuse readers. Alternatively, justify in the Materials and Methods that they are the synonyms.
In this manuscript NG-MAST types and STs are synonyms. We specified this in the Materials and Methods (line 370):
“Allele numbers for the porB and tbpB sequences and sequence types (STs) were assigned via the NG-MAST database (www.ng-mast.net). In this paper, NG-MAST types were also named as STs.”
Line 114: Misprint: 1993, not 193.
The mistake was corrected
“193” was changed to “1993” (Line 121)
Line 119: In reference [19] strains were isolated in 2013-2018, not 2018-2019 as in the text.
Line 128 “In 2018-2019, “ was changed to “In 2013-2018,”
Line 125: What about other isolates? Are they have any genogroup? Describe in the text.
The following sentences were added to the text:
Lines 123-125: “Table 1 represents genogroups containing four or more isolates, thus, 146 out of 183 isolates belonged to the listed genogroups (the rest isolates were not attributed to any genogroup).”
Line 125: "G1993* Corresponds to the European genogroup G1993 [25]". Is this the same genogroup, isn't it?
Right, this is the same genogroup. The text in the Table 1 “Corresponds to the European genogroup G1993” was changed to “European genogroup G1993”.
Line 125: "Two isolates from this genogroup demonstrated decreased sensitivity to ceftriaxone (MICcro = 0.12 mg/L)". The same repeated in the text of the manuscript (lines 130-132).
The text "Two isolates from this genogroup demonstrated decreased sensitivity to ceftriaxone (MICcro = 0.12 mg/L)" was removed from Table 1.
Line 125: "Corresponds to the European genogroup G387 [23]". Maybe reference 25?
Reference in Table 1 was corrected. Due of the addition of three new references, ref. [25] became ref. [28].
Line 256: N.
“N gonorrhoeae” was changed to “N. gonorrhoeae” (line 266)
Line 328: Fix the broken weblink.
The link was corrected (lines 339-340): https://www.eucast.org/fileadmin/src/media/PDFs/EUCAST_files/Breakpoint_tables/v_10.0_Breakpoint_Tables.pdf
Lines 360-364: Please, provide here the reference for the definition of genogroups.
Reference is given:
Line 371 “A phylogenetically related genogroup was defined as described in [28]”
Line 367: Change "RaxML" to "RAxML".
Line 378: RaxML was changed to RAxML/
Line 491: Fix link to the working one: https://www.ecdc.europa.eu/sites/default/files/documents/Molecular-typing-N-gonorrhoeae-web.pdf
The link was fixed (line 512, ref. 28): https://www.ecdc.europa.eu/sites/portal/files/documents/Molecular-typing-N-gonorrhoeae-web.pdf
Reviewer 2 Report
This manuscript by Kandinov et al characterizes 182 isolates of N. gonorrhoeae from several regions in the Russian Federation during the period 2013-2018. The overall goal was to assess the contribution of penA polymorphisms to β-lactam resistance. They find a familiar pattern of β-lactam resistance being associated with mutations in the porB, mtrR, ponA and penA, with TEM-1 observed in some isolates. Encouragingly, no ceftriaxone resistance isolates were found, although two are close to the EUCAST breakpoints. The overall conclusion from the study is that examination of penA sequences in detail is important for predicting resistance levels, beyond NG-MAST genotyping.
Overall, this is an important and rigorously conducted study that highlights the current state of β-lactam resistance among N. gonorrhoeae strains in the Russian Federation and the important contribution of penA mutations. I have only relatively minor comments.
Minor comments
It is insufficient to simply state that the box plots demonstrate correlation between MICs and genetic profile. Some expansion is needed here, e.g. to explain why the CRO MICs for group J are so broad, and similarly for penicillin MICs and group H. Some comment on why the trends for penicillin vs. ceftriaxone MICs differ would be interesting.
I presume group L was excluded from the penicillin box plot due to the TEM-1 plasmid-mediated determinant? Assuming so, this should be mentioned in the text.
Lines 262-263 - It is not clear where Demczuk et al 2020 got their PBP2 numbering from, but three of these five mutations are more correctly referred to as N512Y, A516G and G545S. In fact, to reproduce their numbering here is inconsistent with the numbering in Table 1.
The failure of strain F89 to spread beyond France and Spain is most likely due to a fitness defect and potentially associated with the A501P mutation in PBP2 (penA). By contrast, the FC428 strain has spread to many countries and appears a bigger threat. It would be interesting to discuss the data in light of the penA60 allele found in that strain.
It would also be useful to discuss the data in context of what mutations in PBP2 are known to contribute to cephalosporin resistance through the work of groups in Japan and the US, e.g. i.e. Takahata, S., Senju, N., Osaki, Y., Yoshida, T., and Ida, T. (2006) AAC. 50, 3638 and Tomberg et al (2013) AAC, 57, 3029. I say this because I don’t believe A516G is among the mutations they have identified.
Corrections
Line 43 – define ST at first mention
Lines 45-49 – penA is mentioned twice in this sentence
Line 46 – explain that porB encodes the porin protein PorB1b
Lines 50-52- statement needs a supporting reference
Line 53 – Is it really 300 nucleotide substitutions? One would not expect all positions in the codon to differ for 60-70 amino acids substitutions.
Line 95 – need space between “widespread” and “occurrence”
Fig. 2 – It is almost impossible to read the typeface for this figure.
Table 2 – A bigger space is needed to separate groups A and B
Figs 3 and 4 – Define the dots – outliers presumably
Line 194 – It is customary to refer to this mutation as the Asp345a insertion.
Line 223 – the p for mosaic penA in Table S3 is not < 0.001.
Line 229 – don’t need “effectively”
Author Response
Reviewer 2:
Minor comments
It is insufficient to simply state that the box plots demonstrate correlation between MICs and genetic profile. Some expansion is needed here, e.g. to explain why the CRO MICs for group J are so broad, and similarly for penicillin MICs and group H. Some comment on why the trends for penicillin vs. ceftriaxone MICs differ would be interesting.
Indeed, the scatter of experimental data for ceftriaxone group J and penicillin group H was larger as compared with the other groups. We are sure that the MIC values have been correctly measured. The isolates were divided into groups according to their genetic profile involving the penA allele type and the presence of mutations in genes associated with resistance to β-lactam antibiotics. But one should take into account that there could be other factors effecting the MIC values, for example, non-transformable factor X, which is frequently mentioned in the papers by Magnus Unemo (see, for example Unemo, M.; Shafer, W.M. Antimicrobial resistance in Neisseria gonorrhoeae in the 21st century: past, evolution, and future. Clin Microbiol Rev. 2014, 27, 587–613. doi: 10.1128/CMR.00010-14; Unemo M, Seifert HS, et al. Gonorrhoea. Nat Rev Dis Primers. 2019; 5(1):79. doi: 10.1038/s41572-019-0128-6). Of course, this explanation seems not very good but now we are not able to find any other.
I presume group L was excluded from the penicillin box plot due to the TEM-1 plasmid-mediated determinant? Assuming so, this should be mentioned in the text.
The text was added in lines 210-211:
“(group L with isolates containing blaTEM plasmid was excluded from the penicillin box plot on Figure 4)”
Lines 262-263 - It is not clear where Demczuk et al 2020 got their PBP2 numbering from, but three of these five mutations are more correctly referred to as N512Y, A516G and G545S. In fact, to reproduce their numbering here is inconsistent with the numbering in Table 1.
The amino acid numbering in PBP2 in the paper by Demczuk, W. et al. (Antimicrob Agents Chemother. 2020, 64, e02005-19. doi:10.1128/AAC.02005-19) is shifted by 1 for the amino acids 512 and further. In this manuscript we used the numeration by Magnus Unemo (Unemo, M.; Shafer, W.M. Clin Microbiol Rev. 2014, 27, 587–613. doi: 10.1128/CMR.00010-14). Thus, it is more correct to designate the mutations described in the paper by Demczuk as Asn512Tyr, Ala516Gly and Gly545Ser. In this case they are consistent with the numbering in Table 1.
The text in lines 271-275 was changed to
“..The main determinants affecting MICcro have been shown to be five amino acid substitutions in PBP2 Ala311Val, Ala501Pro/Thr/Val, Asn512Tyr, Ala516Gly, and Gly542Ser (numeration of amino acids Asn513Tyr, Ala517Gly, and Gly543Ser is given in the paper), mutation of the mtrR promoter, substitutions of the Gly120 residue in the PorB protein, and the Leu421Pro substitution in PBP1 [32].
The failure of strain F89 to spread beyond France and Spain is most likely due to a fitness defect and potentially associated with the A501P mutation in PBP2 (penA). By contrast, the FC428 strain has spread to many countries and appears a bigger threat. It would be interesting to discuss the data in light of the penA60 allele found in that strain.
The fact that you are mentioning is undoubtedly very interesting. However, our results do not include investigation of bacterial fitness for individual N. gonorrhoeae isolates. We are only planning such a study. Therefore, we are not sure that it would be correct to discuss the effect of penA allele types on the fitness in this paper.
It would also be useful to discuss the data in context of what mutations in PBP2 are known to contribute to cephalosporin resistance through the work of groups in Japan and the US, e.g. i.e. Takahata, S., Senju, N., Osaki, Y., Yoshida, T., and Ida, T. (2006) AAC. 50, 3638 and Tomberg et al (2013) AAC, 57, 3029. I say this because I don’t believe A516G is among the mutations they have identified.
The Ala516Gly mutation was described in the work by Powell, Tomberg et al (Crystal structures of penicillin-binding protein 2 from penicillin-susceptible and -resistant strains of Neisseria gonorrhoeae reveal an unexpectedly subtle mechanism for antibiotic resistance. J Biol Chem. 2009, 284, 1202–1212. doi: 10.1074/jbc.M805761200 (ref. [26]) who solved the crystal structure of PBP2 obtained from the penicillin-resistant strain FA6140. It was proposed that this mutation at the C-terminal end of PBP2 can reduce the rate of penicillin binding. In addition, linear regression analysis carried out by Demczuk et al (Antimicrob Agents Chemother. 2020, 64, e02005-19. doi:10.1128/AAC.02005-19, ref. [32]) revealed five amino acid substitutions in PBP2 Ala311Val, Ala501Pro/Thr/Val, Asn512Tyr, Ala516Gly, and Gly542Ser that had a largest effect on increasing the ceftriaxone MIC.
Corrections
Line 43 – define ST at first mention
ST was defined at line 47:
“the most frequently identified sequence type (ST) in European countries..”
Lines 45-49 – penA is mentioned twice in this sentence
Line 46 – explain that porB encodes the porin protein PorB1b
The text was changed to (lines 46-52):
“N. gonorrhoeae isolates showing resistance to cephalosporins have been found in many countries worldwide [2–10]. NG-MAST type 1407, the most frequently identified sequence type (ST) in European countries, can develop resistance to third-generation cephalosporins via multiple mechanisms, including, in most cases, the presence of a mosaic allele of the penA gene encoding penicillin-binding protein 2 (PBP2), nucleotide substitutions in the porB gene encoding the porin protein PorB1b, impeding cellular entry of the antibiotic, and a deletion in the promoter of the pump regulator mtrR gene, leading to increased antibiotic efflux [2,4,5,11,12].”
Lines 50-52- statement needs a supporting reference
The necessary references were added (lines 53-55):
“Namely, the mosaic structure of the penA gene, which resulted from homologous recombination between commensal Neisseria species such as N. perflava, N. sicca, N. polysaccharea, N. cinerea, and N. flavescens, is found in numerous cephalosporin-resistant isolates [13-15]”.
13 Ameyama, S.; Onodera, S.; Takahata, M.; Minami, S.; Maki, N.; Endo, K.; Goto, H.; Suzuki, H.; Oishi, Y. Mosaic-like structure of penicillin binding protein 2 gene (penA) in clinical isolates of Neisseria gonorrhoeae with reduced susceptibility to cefixime. Antimicrob Agents Chemother. 2002, 46, 3744–3749. doi: 10.1128/AAC.46.12.3744-3749.2002
- Ito, M.; Deguchi, T.; Mizutani, KS.; Yasuda, M.; Yokoi, S.; Ito, S.; Takahashi, Y.; Ishihara, S.; Kawamura, Y.; Ezaki, T. Emergence and spread of Neisseria gonorrhoeae clinical isolates harboring mosaic-like structure of penicillin-binding protein 2 in central Japan. Antimicrob Agents Chemother. 2005, 49, 137–143. doi:10.1128/AAC.49.1.137-143.2005.
- Osaka, K.; Takakura, T.; Narukawa, K.; Takahata, M.; Endo, K.; Kiyota, H.; Onodera, S. Analysis of amino acid sequences of penicillin-binding protein 2 in clinical isolates of Neisseria gonorrhoeae with reduced susceptibility to cefixime and ceftriaxone. J Infect Chemother. 2008, 14, 195–203. doi:10.1007/s10156-008-0610-7.
Since three new references were added, the reference numbering was shifted by 3 points beginning from ref. 13.
Line 53 – Is it really 300 nucleotide substitutions? One would not expect all positions in the codon to differ for 60-70 amino acids substitutions.
The text was changed to (lines 55-59):
“A mosaic gene can contain multiple nucleotide substitutions that can lead to changes in the amino acid sequence of the protein. For example, mosaic allele 37.001 contains 230 nucleotide substitutions as compared with wild-type penA allele 100.001 that results in the change of 121 amino acids (https://ngstar.canada.ca/alleles/penA)“
Line 95 – need space between “widespread” and “occurrence”
The space was added (Line 102).
Fig. 2 – It is almost impossible to read the typeface for this figure.
Fig. 2 was corrected. But the final formatting will be done by the Editorial office. In addition, all Figures with high resolution were also submitted to the journal in pdf format.
Table 2 – A bigger space is needed to separate groups A and B
We tried to add more space between the groups in Table 2 but the final formatting will be done by the Editorial office.
Figs 3 and 4 – Define the dots – outliers presumably
The box plots contain all the experimental data.
The text was added to the caption of Figure 3
“Data beyond the end of the whiskers are outliers and plotted as points”
Line 194 – It is customary to refer to this mutation as the Asp345a insertion.
ins345 was corrected to ins345a throughout the text (lines 102, 203, 243, 346).
Line 223 – the p for mosaic penA in Table S3 is not < 0.001.
Table S3 (Supplementary material). We do not see the mistake.
The p value for mosaic penA allele XXXIV is 0.0017**, that is < 0.01
***p < 0.001, **p < 0.01,*p < 0.05.
Line 229 – don’t need “effectively”
“effectively” was removed (line 239):
“…β-lactamase, which cleaves penicillins…”